# Investigation of Tumor Heterogeneity Using Integrated Single-Cell RNA Sequence Analysis to Focus on Genes Related to Breast Cancer-, EMT-, CSC-, and Metastasis-Related Markers in Patients with HER2-Positive Breast Cancer

**DOI:** 10.3390/cells12182286

**Published:** 2023-09-15

**Authors:** Sho Shiino, Momoko Tokura, Jun Nakayama, Masayuki Yoshida, Akihiko Suto, Yusuke Yamamoto

**Affiliations:** 1Department of Breast Surgery, National Cancer Center Hospital, Tokyo 104-0045, Japan; asuto@ncc.go.jp; 2Laboratory of Integrative Oncology, National Cancer Center Research Institute, Tokyo 104-0045, Japan; mtokura@ncc.go.jp (M.T.); junakaya@ncc.go.jp (J.N.); 3Department of Diagnostic Pathology, National Cancer Center Hospital, Tokyo 104-0045, Japan; masayosh@ncc.go.jp

**Keywords:** breast cancer, human epidermal growth factor receptor 2, heterogeneity

## Abstract

Human epidermal growth factor receptor 2 (HER2) protein, which is characterized by the amplification of *ERBB2*, is a molecular target for HER2-overexpressing breast cancer. Many targeted HER2 strategies have been well developed thus far. Furthermore, intratumoral heterogeneity in HER2 cases has been observed with immunohistochemical staining and has been considered one of the reasons for drug resistance. Therefore, we conducted an integrated analysis of the breast cancer single-cell gene expression data for HER2-positive breast cancer cases from both scRNA-seq data from public datasets and data from our cohort and compared them with those for luminal breast cancer datasets. In our results, heterogeneous distribution of the expression of breast cancer-related genes (*ESR1*, *PGR*, *ERBB2*, and *MKI67*) was observed. Various gene expression levels differed at the single-cell level between the *ERBB2*-high group and *ERBB2*-low group. Moreover, molecular functions and *ERBB2* expression levels differed between estrogen receptor (ER)-positive and ER-negative HER2 cases. Additionally, the gene expression levels of typical breast cancer-, CSC-, EMT-, and metastasis-related markers were also different across each patient. These results suggest that diversity in gene expression could occur not only in the presence of *ERBB2* expression and ER status but also in the molecular characteristics of each patient.

## 1. Introduction

Breast cancer is the most frequently diagnosed cancer in women worldwide. In particular, for patients with invasive breast cancer, the statuses of hormone receptor (HR: estrogen receptor (ER) or progesterone receptor (PR))) and human epidermal growth factor receptor 2 (HER2) status using immunohistochemical (IHC) staining are the most helpful markers for the treatment decisions of patients with breast cancer in clinical practice. HER2 protein is encoded by the *ERBB2* gene and is a molecular target for HER2-overexpressing breast cancer. Many targeted HER2 therapies have been well established thus far. Although many types of targeted therapy markedly improve survival for patients with HER2-positive breast cancer, it still remains the cause of recurrence or death in those patients [1].

Intratumor heterogeneity in breast cancer has been frequently observed in morphological and biomarker patterns. The heterogeneity of HER2 expression can be observed for HER2-positive breast cancer with immunohistochemical (IHC) staining. Two types of heterogeneity are suggested, such as two distinct populations or continuous transitions [2]. Importantly, the heterogeneity of HER2 is considered one of the mechanisms for drug resistance [1], and HER2 heterogeneity could be associated with worse outcomes compared with nonheterogeneous groups [3]. The therapeutic response to neoadjuvant therapy for HER2-positive breast cancer is different between the heterogeneous group and the nonheterogeneous group [4]. However, the molecular mechanisms of heterogeneity of HER2 have not been fully elucidated.

Traditional bulk RNA sequence analysis is helpful to understand gene expression in tumor tissues, but it could be limited for comparing gene expression among each cell. This is because this method captures only an “average” of the expression profiles of total cells. However, single-cell RNA-seq (scRNA-seq) is a powerful technique to investigate gene expression in each single cell and to understand tumor heterogeneity. Nevertheless, performing this technique on larger cases could be limited due to high sequencing costs. Thus, it might be difficult to statistically analyze limited numbers. Moreover, some articles have suggested that single-cell meta-analysis, which integrates a large-scale single-cell cohort as a meta-analysis technique, is a useful technique to solve this problem [5,6,7]. Therefore, we also performed an integrated analysis of both scRNA-seq data from public databases and our cohort data to investigate the intratumor heterogeneity of HER2-positive breast cancer.

This study aimed to investigate intratumor heterogeneity in HER2-positive breast cancer cases by comparison with luminal breast cancer (HR-positive/HER2-negative) using an integrated scRNA-seq cohort because we aimed to focus on the comparison between the luminal and HER2 subtypes to investigate the relationship between ER and HER2 status [8]. We also performed clustering analysis for all tumor cells and identified differential gene expression in each classified cluster, particularly focusing on genes related to cancer stem cells (CSCs), epithelial-to-mesenchymal transition (EMT), and metastasis. Additionally, we conducted pathway analysis for a set of genes involved in the expression of *ERBB2*.

## 2. Materials and Methods

### 2.1. Data Collection

The scRNA-seq cohorts were downloaded from the public Gene Expression Omnibus (GEO) database (Appendix A). We selected the scRNA-seq cohorts, which could be downloaded from the public GEO database and analyzed with a Chromium platform (10× Genomics, CA, USA). Moreover, we used data sets with publicly available clinicopathological information, such as patient age, tumor size, and breast cancer subtype. Regarding breast cancer subtype, it is necessary to obtain information on HR (either ER or PR) and HER2 IHC score or HER2 amplification status. The definition of each breast cancer subtype is as follows.

Luminal subtype: Cases with either ER or PgR positivity on IHC staining. ER and PgR status were considered positive using a cut off value of ≥ 1% according to the American Society of Clinical Oncology/College American Pathologists (ASCO/CAP) guideline [9]. We considered either 2+ or 3+ as positive in datasets which were described as 0+, 1+, 2+, or 3+.HER2 subtype: Cases with HER2 positivity. We considered 3+ or 2+ with HER2 amplification on IHC staining as positive with reference to the ASCO/CAP guideline [10]. The results of HER2 amplification is necessary in cases with unavailable HER2 results on IHC score.

From scRNA-seq profiling of breast cancer, *BRCA1* mutant preneoplastic mammary gland cell and normal mammary gland cell studies, six HER2 amplified samples were extracted from a total of 69 samples in the GSE161529 dataset [11]. Four HER2 amplified samples and eight luminal samples were extracted from a total of 26 samples in the GSE176078 dataset [12], and one luminal and two HER2 amplified samples were extracted from a total of 15 samples in the GSE180286 dataset [13]. Two HER2-positive and four luminal samples of scRNA-seq analysis were performed at our institution [14]. A total of 27 samples (HER2 type: 14 cases; luminal type: 13 cases) were collected, and the details of the extracted samples are shown in Appendix A. These datasets were imported into R software version 4.2.0 (R Development Core Team) and transformed into Seurat objects with the package Seurat version 4.3.0 [15]. The Seurat objects from the different datasets were then integrated in R.

### 2.2. The Integration of Datasets, Data Quality Control, and the Removal of Batch Effects

The integrated dataset was subjected to normalization, scaling, and principal component analysis (PCA) with Seurat functions. Low-quality cells were removed from the merged dataset before batch effect removal according to the following criteria: nFeature_RNA > 500 and percent.mt < 20. The expression counts of each sample were normalized by log normalization with the function “NormalizeData” in Seurat. Doublet cells in the integrated dataset were removed by the DoubletFinder method version 2.0.3. [16,17]. To remove the batch effect between cohort studies, Harmony version 0.1.1. algorithms were applied to the integrated datasets [18,19] following the instructions in the Quick start vignettes (https://portals.broadinstitute.org/harmony/articles/quickstart.html, accessed on 1 February 2023).

### 2.3. Data Clustering and Cell Type Annotation

The clustering of neighboring cells was performed by the functions “FindNeighbors” and “FindClusters” from Seurat using Harmony reduction. First, the clusters were grouped based on the expression of tissue compartment markers (for example, *KRT8* for epithelia, *CLDN5* for endothelia, *COL1A2* for fibroblasts, and *PTPRC* for immune cells) (Figure 1) and then annotated according to “A molecular cell atlas of the human breast” [14,20].

### 2.4. Pathway Enrichment Analysis

We performed enrichment analysis with the signature gene list from each epithelial cell cluster using the “ClusterProfiler” packages in R [21]. For enrichment analysis, gene symbols were converted to ENTREZ IDs using the “org.Hs.eg.db” package (Carison M, R package version 3.10.0., 2019). GO enrichment analysis using the “enrichGO” function was performed using the BH method.

### 2.5. Data Visualization

General UMAP plots, feature plots, and violin plots were generated by Seurat in R. Box plots and bubble plots were generated using the “ggplot2” package in R. Scatter plots with lines were generated by Excel software (Microsoft, Excel 2019 MSO). Heatmaps were generated based on Pearson correlation by the online software Morpheus (https://software.broadinstitute.org/morpheus/, accessed on 1 August 2023). We selected representative EMT markers and CSC markers based on previous reports [22,23].

## 3. Results

### 3.1. Integrated Single-Cell RNA-seq Data for 14 HER2 Subtypes and 13 Luminal Subtypes

We integrated scRNA-seq data of patients with breast cancer, consisting of 3 publicly available datasets (GSE161529, GSE176078, GSE180286) and our own datasets (GSE195861) (Figure 1A and Appendix A). After the process of quality control using Seurat, a total of 113,135 single cells were obtained from 27 scRNA-seq datasets (HER2-positive, 14 cases: 77,792 samples; luminal, 13 cases: 35,343 samples). Clinicopathological data on each case are listed in Appendix A. The median age for total patients was 53 years old (range: 32–80). The median cell number for each case was 3375 (range: 345–14,224). Clustering analysis using UMAP plots revealed 27 clusters (Figure 1B). Based on the expression of cell-type-specific markers, we classified 12 typical clusters (Figure 1C). They were largely segregated into four groups by specific markers, as follows: epithelial cells (*KRT8*), immune cells (*PTPRC*), fibroblast cells (*COL1A2*), and endothelial cells (*CLDN5*) (Figure 1D). The proportions of those 12 cell classifications for each subtype are shown in Figure 1E. In both HER2 and luminal subtypes, epithelial cells accounted for much of the cell population. There was no statistically significant difference for each proportion of cell classification between those two subtypes.

### 3.2. The Intra-Tumor Heterogeneity of the Epithelial Population in the HER2 and Luminal Subtype

We extracted the clusters of epithelial cells for further analysis and reconstructed the UMAP plots, as shown in Figure 2A for classifying into new clusters and Figure 2B for showing each subtype. A total of 51,476 cells were collected, and 11 new clusters were revealed in the UMAP plots. The UMAP plots with specific markers (*ESR1*, *PGR*, and *ERBB2* as breast cancer markers, *KRT5* as a basal marker, *ACTA2* as a myoepithelial marker, and *MKI67* as a proliferation marker) are shown in Figure 2C. The expression of *ESR1* and *ERBB2* was diffusely distributed on each cluster but the intensities were different among each cluster (Figure 2D). Meanwhile, the expression of *PGR* was low on most clusters, and the expression levels of *KRT5* and *ACTA2* were basically low but high in cluster 8. High expression of *MKI67* was observed in cluster 5, indicating a proliferating epithelial population.

The proportion of epithelial cells per cluster is shown in Figure 3A. In particular, the proportion of cell number on cluster 3 was observed to be statistically higher in the luminal subtype compared with the HER2 subtype (*p* < 0.05) (Figure 3B). Meanwhile, the proportion of cluster 7 was higher in the HER2 subtype compared with the luminal subtype, although the difference was marginally different (*p* = 0.065). The remaining clusters showed no significant differences. We then performed pathway enrichment analysis based on the clusters to identify cluster-specific functions and signatures (Figure 3C). The function of cluster 1 was identified as correlating with estrogen-dependent gene expression. Cluster 5 was associated with cell proliferation, which was compatible with high *MKI67* expression in the cluster analysis. Meanwhile, cluster 7, which was up regulated in the HER2 subtype, was associated with cell adhesion. Clusters 3 and 9 might be associated with immune response based on the GO terms. Further, clusters 2, 4, 6, and 10 contained similar GO terms, such as peptide chain elongation and eukaryotic translation elongation; thus, they could be similar cell populations, although they were classified as different clusters.

### 3.3. The Expression Levels of Typical Breast Cancer-, CSC-, EMT-, and Metastasis-Related Markers across Patients

We also examined typical gene expressions which are associated with CSC, EMT, and metastasis in each case (Figure 4). The expression of *ESR1* and *ERBB2* was largely distinct across the cases. The *ESR1* expression of the luminal-4 case was the highest in all cases. Meanwhile, the *ERBB2* expression was higher in HER2-3, 6, 7, 8, 10, and 11. The expression of *EGF* in the luminal-4 case was evidently the highest among all the cases. Higher expression levels of EMT markers were especially observed in HER2-2, -5, -6, -10, and -11 and luminal-3, -6, -11, and -12. In those cases, the expression of breast cancer markers, such as *ESR1* and *ERBB2*, was relatively low in the HER2-2 and luminal-3, -6, and -11 cases. A high expression of CSC markers was partially observed in HER2-2, -5, -6, and -8 and luminal-5 and -6. The expression of CSC markers in HER2 cases slightly tended to be higher compared with luminal cases, but this was not statistically different. Meanwhile, the expression of *MKI67* was higher in HER2-11 and 14 and luinal-11, but had no obvious correlation with breast cancer-, CSC-, EMT-, and metastasis-related markers.

In Figure 5, we investigated the correlation between breast cancer markers, such as *ESR1* and *ERBB2*, and EMT markers, since we considered that the expression of *ESR1* and *ERBB2* tended to reduce in the cases with high expression of EMT in Figure 4. The relationship between the expression of *ERBB2* and *TJP1* found a moderate correlation (R^2^ = 0.3822), as shown in Figure 5A, but the correlation between *ERBB2* and the other EMT markers was low. In contrast, the expression levels of *ESR1* were inversely correlated with EMT markers (Figure 5B). We also investigated the correlation between EMT-related transcription factors and each breast cancer marker (Figure 5C,D). The expression of *ESR1* and transcription factors was totally inversely correlated, but the expression of *ERBB2* and transcription factors had no correlation in HER2-type breast cancer. These data suggested that *ESR1* negatively regulated the EMT process but not *ERBB2*.

### 3.4. The Expression Levels of Typical Breast Cancer-, CSC-, EMT-, and Metastasis-Related Markers across the Epithelial Clusters

We investigated the differential expression of breast cancer-, CSC-, EMT-, and metastasis-related markers between HER2 and luminal subtypes on each epithelial cluster (Figure 6). We mainly focused on clusters 1–7 because the number of cells on clusters 8–11 was slightly smaller (Appendix A). As for cluster 5, the expression of *CDH2* was evidently higher in the HER2 subtype, but there were no significant differences in the transcription factors overall. The expression levels of *SOX2* was higher in the HER2 cases in all clusters. The expression level of *SNAI2* in cluster 4 and *CD36* in cluster 2 were slightly higher in the luminal cases. The expression of *MKI67* in cluster 6 was higher in HER2 case compared with the luminal case. 

### 3.5. Comparison of Gene Expression between the ERBB2-High Group and ERBB2-Low Group

Focusing on only HER2 cases, we compared the differences in gene expression based on the average *ERBB2* expression. We distributed the *ERBB2*-high group into the *ERBB2*-low group based on the average *ERBB2* expression in all cases and conducted a pathway analysis. The UMAP plots of both the *ERBB2*-high group and the *ERBB2*-low group are shown in Figure 7A. The pathway analysis revealed that upstream regulators included cell proliferation factors, such as *IGF*, *IGF1R*, and *EGF*, and immune related factors (Figure 7B). Moreover, 12 miRNAs were predicted to be upstream regulators of *ERBB2* expression (Appendix A).

### 3.6. Comparison of the Gene Expression Levels of Typical Breast-Cancer-Related Markers on Each Subtype at Single-Cell Levels: Luminal-HER2 Subtype, Pure-HER2 Subtype, and Luminal Subtype

We investigated the gene expression levels of typical breast-cancer-related markers (*ESR1*, *PGR*, *ERBB2*, and *MKI67*) on the two subtypes (luminal-HER2 subtypes and pure-HER2 subtypes) compared with the luminal subtypes (Appendix A: luminal-HER2 subtypes, Appendix A: pure-HER2 subtypes, and Appendix A: luminal subtypes). The expression of *ERBB2* was diffusely distributed on each cluster of pure-HER2 subtype but was significantly higher in one of clusters in the luminal-HER2 subtype. High *MKI67* expression was evidently identified in one of the clusters on all subtypes.

The expression of *ESR1* was relatively higher in the luminal subtype compared with the luminal-HER2 subtypes (Appendix A). The pure-HER2 subtypes clearly had lower expression of *ESR1* compared with the other subtypes, but some expression was still observed. The expression of *ERBB2* was more elevated in the pure-HER2 subtypes compared with the luminal-HER2 subtype. The luminal subtypes had lower *ERBB2* expression but had a slight amount of expression, even though the HER2 results were defined as non-amplified in the pathological findings.

We compared gene expressions between the luminal-HER2 and pure-HER2 subtypes and conducted pathway analysis to investigate the influence of ER status in HER2-positive breast cancer. The HER2 subtypes in our dataset included both the luminal-HER2 subtypes (ER-positive/HER2-positive cases: CID3586 and p2) and pure-HER2 subtypes (ER-negative/HER2-positive cases: CID3921, CID45171, CID3838, p3, NCCJN1, and NCCJN2). The UMAP plots of both the luminal-HER2 subtypes and pure-HER2 subtypes are shown in Figure 8A. The pathway analysis revealed that multiple genes including *ESR1*, *CTNNB1*, and *MYC* were predicted as upstream regulators (Figure 8B). Moreover, 44 miRNAs were predicted to be upstream regulators (Appendix A).

## 4. Discussion

We investigated intratumoral genetic heterogeneity in scRNA-seq datasets of HER2-positive breast cancer compared with luminal breast cancer. There was almost the same proportion of cell classification in the clustering analysis between the two datasets. Moreover, we compared the difference in molecular function between the *ERBB2*-high group and *ERBB2*-low group based on the average *ERBB2* expression for total cases and identified upstream regulators based on *ERBB2* expression levels. We also identified the difference in the molecular functions between the luminal-HER2 subtype and the pure-HER2 subtype at the single-cell level. Furthermore, the gene expression levels of typical breast cancer-, CSC-, EMT-, and metastasis-related markers were also different across each patient. EMT-, CSC-, and metastasis-related markers were markedly higher in some cases, and the expression of *MKI67* was not correlated with those markers.

Clustering analysis revealed that HER2 cases had relatively higher expression levels of genes related to cell adherence pathways, such as smooth muscle contraction or cell junction organization, compared with luminal cases in pathway enrichment analysis, and we identified some genes, such as *Rac3* or *Rab32* (data not shown), that were related to cell adhesion [24,25]. These genes have been reported as biomarkers associated with the EMT. The EMT plays an important key role in metastasis and is also associated with chemotherapy resistance [26]. Ingthorsson. S, et al. reported that the amplification of *ERBB2* could lead to EMT and tumorigenicity [27], and our results for HER2-positive cases were compatible with those of previous reports. Interestingly, EMT markers also had higher expression in some luminal cases. Thus, the expression of EMT markers could be elevated not only in HER2 cases but also in luminal cases. This could mean that the biological features of each case, such as the expression of EMT markers, vary regardless of HER2-positive cases and luminal cases.

*MKI67* is considered an important biomarker for classifying breast cancer. The expression of *MKI67* varied in each case, and some cases had markedly higher expression than other cases. Moreover, *MKI67* was not correlated with either EMT markers or CSC markers in each case. Therefore, clinicians might need to consider treatment strategies focusing on not only *MKI67* markers but also EMT- or CSC-related markers.

The *ERBB2* expression levels in HER2 datasets were quite varied in each epithelial cell. Furthermore, we found that molecular function could differ between the *ERBB2*-high group and *ERBB2*-low group in pathway analysis. In particular, some proliferation factors, such as *IGF1* and *EGF*, were identified as upstream regulators in the *ERBB2*-high group. Crosstalk between *IGF* signaling and *ERBB2* has been reported thus far and could lead to anti-HER2 therapy resistance [28]. Some trials reported that patients with high *ERBB2* mRNA expression levels had a high pCR rate compared with those with low levels [4,29,30]. Therefore, our results may provide clues to elucidate the molecular mechanisms involved in predicting the response to anti-HER2-targeted therapy. Additionally, we found immune-related factors using pathway analysis according to *ERBB2* expression levels. Many previous studies have shown that TILs are associated with the HER-enriched type [31,32], and our results were also the same.

A previous meta-analysis suggested that the response to drug treatment and pathological features were different between luminal-HER2 breast cancers and pure-HER2 breast cancers [29]. In our results, the average *ERBB2* expression levels were higher in the pure-HER2 subtype than in the luminal-HER2 subtype. Interestingly, the expression of *ERBB2* was distributed on all clusters in the pure-HER2 subtype but was strongly focused on one of the clusters in the luminal-HER2 subtype. These findings could suggest that the luminal-HER2 subtype could be more heterogeneous than the pure-HER2 subtype. Filho M et al. reported that the number of cases defined as having HER2 heterogeneity was significantly higher in IHC level 2+ than in IHC level 3+ [4]. Therefore, our results also suggest that the diversity of *ERBB2* mRNA expression could be strongly associated with HER2 heterogeneity.

In our pathway analysis, we found molecular differences between the luminal-HER2 subtype and the pure-HER2 subtype. *MYC* expression levels were identified as upstream regulators in luminal-HER2 cases compared with pure-HER2 cases. This could mean that *MYC* expression in pure-HER2 cases at the single-cell level was higher than that in luminal-HER2 cases. *MYC* has been reported to promote cell proliferation in breast cancer [33], and HER2 overexpression could lead to *MYC* amplification [34]. Therefore, our results suggest that pure-HER2 cases could be more aggressive than luminal-HER2-positive cases. Further investigations are needed to analyze the target *MYC* expression levels in each breast cancer subtype.

We showed that *ERBB2* was slightly expressed in patients who were clinically determined to be luminal cases in our datasets. Luminal datasets included a variety of cases, either 0, 1+, or 2+ (nonamplification) on IHC results. IHC scores of 1+ or 2+ breast cancer have been classified as HER2-low breast cancer [35], and *ERBB2*-targeted therapy (trastuzumab deruxetecan) showed prolonged survival in patients with HER2-low breast cancer [36]. Additionally, *ERBB2* expression levels were significantly higher in HER2-low breast cancers than in HER2 0 [37]. As our datasets for the luminal subtype included cases with 1+ and 2+ IHC results, slight expression of *ERBB2* might be identified in luminal datasets. Further investigations of gene differences at the single-cell level between HER2 0 and HER2-low breast cancer could have more beneficial implications in clinical practice.

The scRNA-seq process has huge potential to uncover tumor heterogeneity, but some studies emphasized challenges in the technique [38,39]. First, scRNA-seq analysis tools might be insufficient to accurately represent the number of cell populations and gene expression insights of capturing efficiency or cell viability. Therefore, we focused on genes with higher expression rather than those with low expression by removing low-quality cells. Next, the accuracy of each scRNA-seq datum depends on the experience of each facility. Our integrated scRNA-seq results may demonstrate selection bias, although all datasets have already been published thus far and a certain level of reliability has been ensured.

In addition, scRNA-seq analysis provides a large amount of sequencing data; thus, studies are needed to carefully interpret such results, considering treatment approaches and effects on patient prognosis. First, our results identified intra-tumoral heterogeneity, but interpatient heterogeneity could also be important along with that issue, as shown in Figure 4. Therefore, clinicians need to consider medical treatment not only focusing on the genetic differences in each tumor cell but also developing treatment, following the genetic characteristics of each patient in the future. Next, the distribution of *ERBB2* expression significantly differed among three subtypes, including HER2, luminal-HER2, and luminal subtypes. *ERBB2* expression was diffusely distributed on all clusters in the pure-HER2 subtype. Meanwhile, *ERBB2* expression was localized in a specified cluster. The results indicate that other clusters with no *ERBB2* expression could have resulted in the absence of response for HER2-targeted therapy. Clinicians may need to carefully consider treatment strategies for breast cancer with the luminal-HER2 subtype rather than the pure-HER2 subtype. Interestingly, slight *ERBB2* expression was found in the luminal subtype, and the results were compatible with the previous studies on clinical trials that *ERBB2*-targeted therapy demonstrated better survival in patients with HER2-low breast cancer [36].

MicroRNAs (miRNAs) are small noncoding RNAs that modulate gene activity by binding to the 3′ untranslated region of a specific gene. Our results found multiple miRNAs that are considered upstream regulators based on *ERBB2* expression levels. Many miRNAs associated with HER2-positive breast cancer have been reported thus far [40]. Some miRNAs were identified as upstream regulators that were upregulated in the *ERBB2*-high group compared with the *ERBB2*-low group and have been reported as cell proliferation markers for breast cancer [41,42]. The *ERBB2*-high group was predicted to have more aggressive disease than the *ERBB2*-low group, and these results were compatible with those of previous studies. Furthermore, miR-1-3p and miR-124-3p are considered to be suppressor genes for breast cancer [43,44]. It could not be clarified why those miRNAs were predicted to be upregulated in the *ERBB2*-high group. The number of upstream miRNAs was higher in the analysis according to ER status (a total of 44 miRNAs) than in the analysis according to *ERBB2* expression (12 miRNAs). These results could suggest that more miRNAs could be associated with ER expression in HER2-positive breast cancer. Moreover, miR-29 has been reported to play a role in the EMT in breast cancer [45] and was predicted to be upregulated in the luminal-HER2 subtype compared with the pure-HER2 subtype in our results. Moreover, some miRNAs were identified as upstream regulators according to both ER status and *ERBB2* expression (Appendix A). Interestingly, let-7 was predicted to be slightly downregulated in the *ERBB2*-high group compared with the *ERBB2*-low group but upregulated in the luminal-HER2 subtype compared with the pure-HER2 subtype. Let-7 contributes to the regulation of several signaling pathways, such as breast cancer cell growth, the regulation of *MYC* expression, and the regulation of CSC properties [46,47], and could be a potential therapeutic target for breast cancer patients.

There were some limitations in our study. First, we integrated single-cell data using some clinical datasets. Therefore, our analytic results could be biased by the molecular characteristics of each case. We need to investigate using more cases of single-cell data to minimize the biased problem. Second, we could not obtain the IHC staining results for each case from the public database. For example, for some cases, information on hormone receptor status could not be obtained, but those cases were included in the HER2 datasets.

## 5. Conclusions

In conclusion, our integrated single-cell study demonstrated intratumoral genetic heterogeneity in cases with HER2-positive breast cancer. Clusters associated with immune response and cell adherence pathways may have increased expression levels in the HER2 subtype. Molecular functions could vary depending on the expression of *ERBB2* and ER status. Moreover, the luminal-HER2 subtype could be more heterogenous compared with pure-HER2. Meanwhile, there could be differences in the gene expression patterns of typical breast cancer-, CSC-, EMT-, and metastasis-related markers across each patient. These results suggest that understanding and elucidating the molecular mechanisms of HER2-positive breast cancer would require the consideration of not only intratumoral heterogeneity but also interpatient heterogeneity.

## Figures and Tables

**Figure 1 cells-12-02286-f001:**
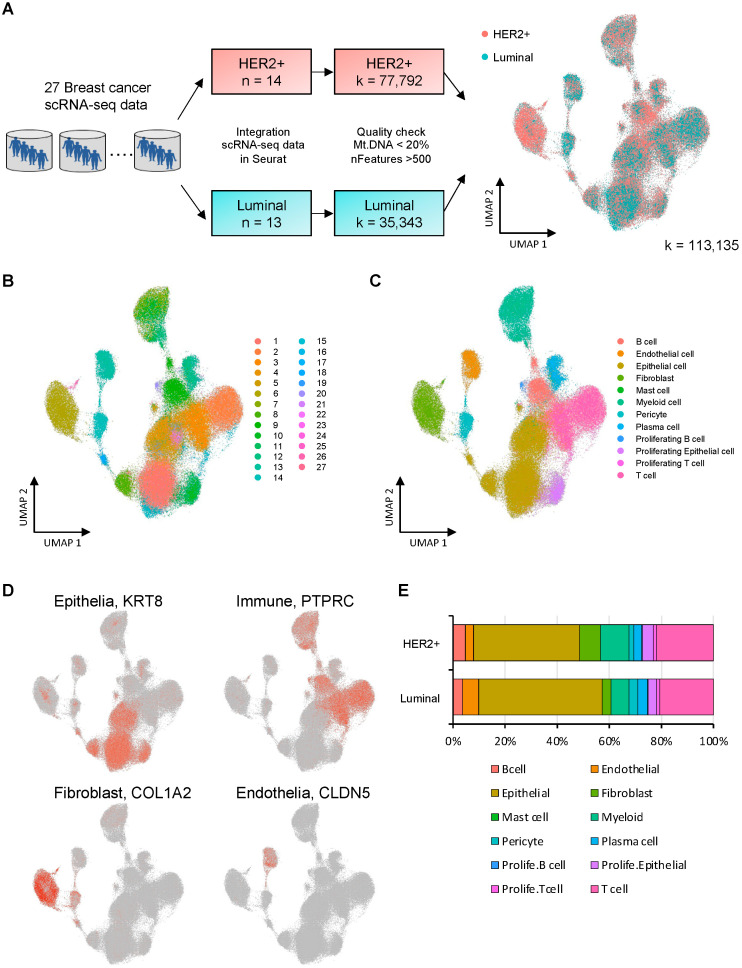
Construction of the mapping of cases with HER2 cases and luminal cases by 27 scRNA-seq datasets (**A**–**E**). (**A**) Flowchart of the construction of integrated single cell analysis. (**B**) A UMAP plot displaying 113,135 single human breast cells. (**C**) A UMAP plot displaying cell populations for the cell type clusters. (**D**) UMAP plots with representative marker expression for the cell type clusters. (**E**) A comparison of cell populations of immune cells, epithelial cells, endothelial cells, and fibroblast cells.

**Figure 2 cells-12-02286-f002:**
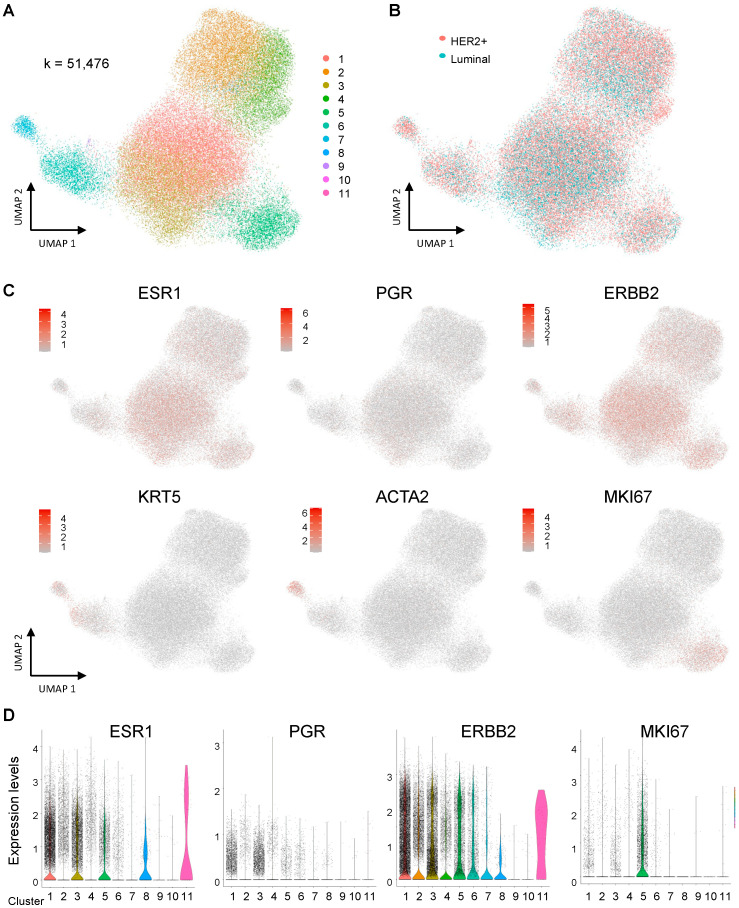
Single-cell analysis of the epithelial components of human breast cancer (**A**–**D**). (**A**) A UMAP plot of epithelial cells including both HER2 and luminal breast samples. (**B**) A UMAP plot on each subtype. (**C**) UMAP plots with representative marker expression for the cell type clusters. (**D**) Violin plots with breast-specific markers (*ESR1*, *PGR*, *ERBB2*, and *MKI67*).

**Figure 3 cells-12-02286-f003:**
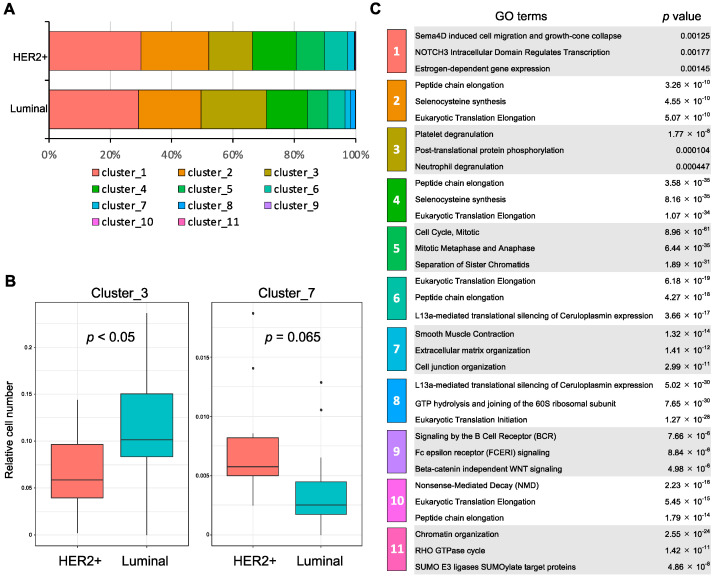
Comparison of the proportion of cells per cluster and pathway enrichment analysis (**A**–**C**). (**A**) Comparison of the proportion of cells between HER2 cases and luminal cases on each cluster. (**B**) The differences in relative cell number on cluster 3 and cluster 7. (**C**) Pathway enrichment analysis of epithelial components based on cluster.

**Figure 4 cells-12-02286-f004:**
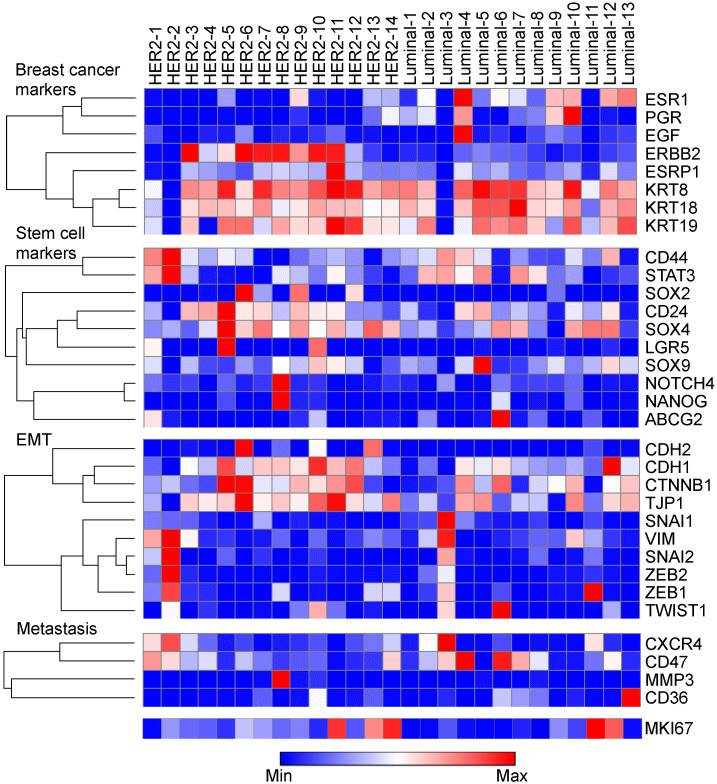
Heatmap of differentially expressed genes in each case.

**Figure 5 cells-12-02286-f005:**
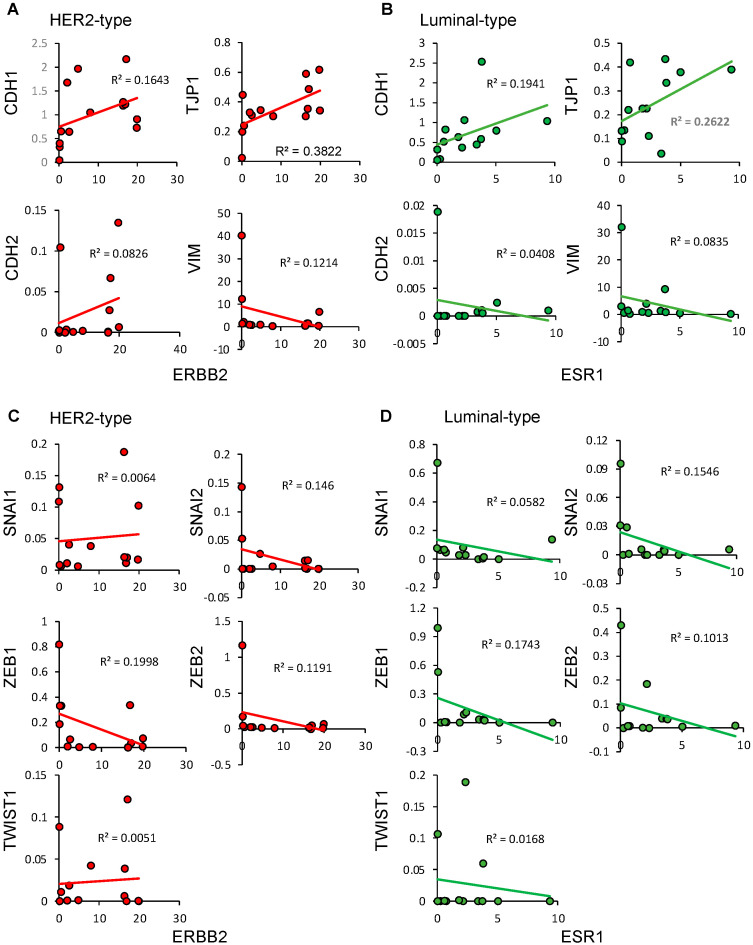
The correlation between breast cancer markers, EMT markers, and EMT-related transcription factors (**A**–**D**). (**A**) The correlation between ERBB2 and EMT markers in the HER2 cases. (**B**) The correlation between ERBB2 and EMT markers in the luminal cases. (**C**) The correlation between ERBB2 and EMT-related transcription factors in the HER2 cases. (**D**) The correlation between ERBB2 and EMT-related transcription factors in the luminal cases.

**Figure 6 cells-12-02286-f006:**
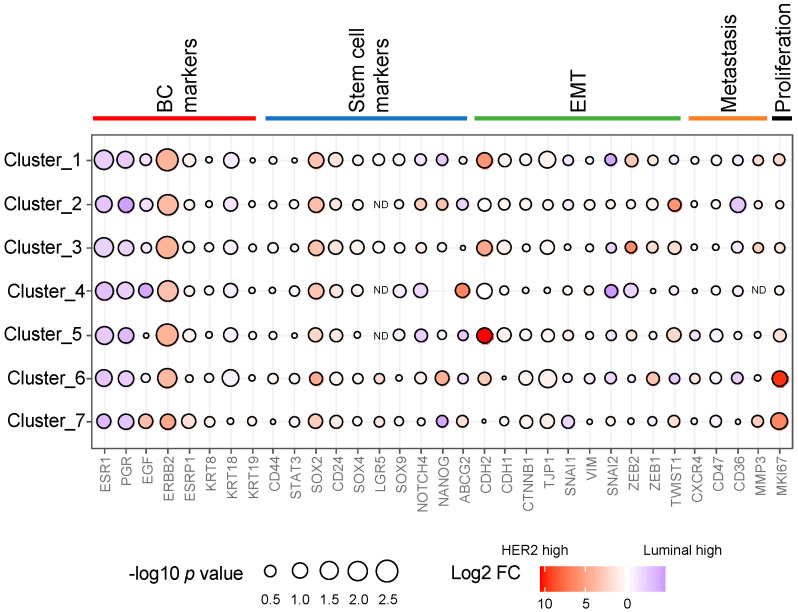
Comparison of gene expression for stem cell markers, EMT, and metastasis by cluster.

**Figure 7 cells-12-02286-f007:**
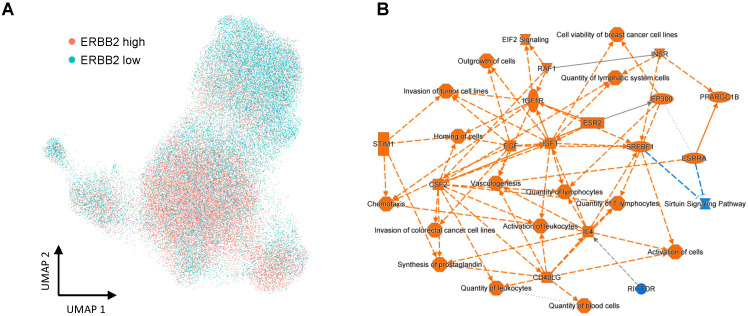
Pathway enrichment 
analysis based on *ERBB2* expression (**A**,**B**). (**A**) 
A UMAP plot of *ERBB2*-high and -low populations. (**B**) Pathway 
enrichment analysis of luminal epithelial components based on *ERBB2* 
expression.

**Figure 8 cells-12-02286-f008:**
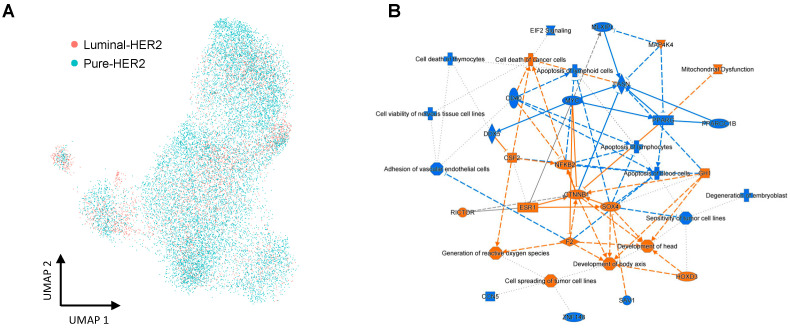
Pathway enrichment analysis based on ER status (**A**,**B**). (**A**) A UMAP plot of 
the luminal-HER2 and pure-HER2 subtypes in HER2+ breast cancer cells. (**B**) 
Pathway enrichment analysis of luminal epithelial components based on ER 
status.

## Data Availability

The scRNA-seq cohorts were downloaded from GSE161529, GSE176078, GSE180286, and GSE195861 in the public Gene Expression Omnibus (GEO).

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
