# Peer review of "Investigation of Tumor Heterogeneity Using Integrated Single-Cell RNA Sequence Analysis to Focus on Genes Related to Breast Cancer-, EMT-, CSC-, and Metastasis-Related Markers in Patients with HER2-Positive Breast Cancer"

_cells, 2023, doi:10.3390/cells12182286_

Round 1

Reviewer 1 Report

 Shiino et al, in this manuscript titled "Investigation of tumor heterogeneity using integrated single- 2 cell RNA sequence analysis to focus on genes related to breast 3 cancer-, EMT-, CSC-, and metastasis-related markers in patients 4 with HER2-positive breast cancer"  have performed an integrated analysis of the breast cancer single-cell gene expression data for HER2-positive breast cancer cases from both scRNA-seq data from public datasets and data from our cohort and compared them with those for luminal breast cancer datasets.

Although the manuscript is scientifically sound, there are a few concerns, which needs to be addressed before its acceptance.

1. Authors should elaborate more on how they have chosen data sets in this work.

2. Sub-section 3.6 and 3.7 could be combined to avoid confusion.

3. Conclusion section should contain significant of the work, which also mention results obtained.

4. Please check for all the grammatical errors throughout the manuscript.

Minor corrections are required.

Author Response

Reviewer #1

Shiino et al, in this manuscript titled "Investigation of tumor heterogeneity using integrated single- 2 cell RNA sequence analysis to focus on genes related to breast 3 cancer-, EMT-, CSC-, and metastasis-related markers in patients 4 with HER2-positive breast cancer"  have performed an integrated analysis of the breast cancer single-cell gene expression data for HER2-positive breast cancer cases from both scRNA-seq data from public datasets and data from our cohort and compared them with those for luminal breast cancer datasets.

Although the manuscript is scientifically sound, there are a few concerns, which needs to be addressed before its acceptance.

  1. Authors should elaborate more on how they have chosen data sets in this work.

A1. We appreciate your suggestion. We selected the scRNA-seq cohorts, which could be downloaded from the public GEO database and analyzed with a Chromium platform (10x Genomics, CA, USA). Moreover, we used data sets with publicly available clinicopathological information, such as patient age, tumor size, and breast cancer subtype. We elaborated more on the data set selection in the Methods section (page 2, lines 80–83).

  1. Sub-section 3.6 and 3.7 could be combined to avoid confusion.

A2. We combined sub-sections 3.6 and 3.7 in the Results section to avoid confusion.

  1. Conclusion section should contain significant of the work, which also mention results obtained.

A3. Thank you for your suggestion. We amended conclusion session according to our insights according to the results as below.

“In conclusion, our integrated single-cell study demonstrated intratumoral genetic heterogeneity in cases with HER2-positive breast cancer was observed in our integrated single-cell study. Clusters associated with immune response and cell adherence pathways may have increased expression levels in the HER2 subtype. Molecular functions could vary depending on the expression of ERBB2 and ER status. Moreover, the Luminal-HER2 subtype could be more heterogenous compared with pure HER2. Meanwhile, there could be differences in the gene expression patterns of typical breast cancer-, CSC-, EMT-, and metastasis-related markers across each patient. These results suggest that understanding and elucidating the molecular mechanisms of HER2-positive breast cancer would require consideration of not only intratumoral heterogeneity but also interpatient heterogeneity.”

  1. Please check for all the grammatical errors throughout the manuscript.

A4. Springer Nature Author Services already checked all sentences in the manuscript. We attached the certification statement to the submission file. Additionally, we have carefully rechecked the revised manuscript.

Reviewer 2 Report

Breast cancer is a highly heterogeneous tumor, both in terms of morphology and biomarker expression. Therefore, researching this heterogeneity and its related clinical significance and prognostic information is currently a focal point of research. This article utilizes data from public single-cell sequencing as well as in-house sequencing data. The volume of single-cell sequencing data is substantial, but how to analyze, present, and derive meaningful insights from the results are questions that the authors of this paper should carefully consider.

By employing bioinformatics tools, cells are classified into different clusters. However, the characteristics of these classifications, their implications for breast cancer management, how to selectively choose treatment approaches, and their effects on patient prognosis should be prominently discussed. This discussion should go beyond the presentation of data, as it holds significant importance.

In addition, there are several issues that need to be addressed:

  1. Figure 2D lacks MKI67 plots.
  2. It is necessary to provide clear definitions for the Luminal subtype and HER2 subtype.
  3. In Figure 3B, cluster 3 and 7 were selected for comparison. However, what about the diffusion across other clusters? In clinical practice, breast cancer is categorized into Luminal A, Luminal B, HER2-positive, and triple-negative subtypes. It's important to define the Luminal and HER2 groups and provide an explanation for their selection.
  4. GO analysis was performed for each cluster. By examining the specific differences in GO terms within each group, what cellular characteristics do these differences indicate, and what insights do these data provide?
  5. The criteria for selecting EMT markers and CSC markers need to be clarified。
  6. There seems to be inconsistency in cluster numbers. Figure 3A shows 11 clusters, while Figure 4 indicates that the HER2 group has 14 clusters and the Luminal group has 13 clusters.

 Minor editing of English language required.

Author Response

Reviewer #2

Breast cancer is a highly heterogeneous tumor, both in terms of morphology and biomarker expression. Therefore, researching this heterogeneity and its related clinical significance and prognostic information is currently a focal point of research. This article utilizes data from public single-cell sequencing as well as in-house sequencing data. The volume of single-cell sequencing data is substantial, but how to analyze, present, and derive meaningful insights from the results are questions that the authors of this paper should carefully consider.

A. We appreciate your invaluable comment. As you suggested, researchers need to carefully consider the scRNA-seq data despite the huge potential of this tool to uncover tumor heterogeneity. Some articles have reviewed the potentials and limitations of scRNA-seq analyses (reference numbers: 38,39). First, scRNA-seq analysis tools might be insufficient to accurately represent the number of cell populations and gene expression insights of capturing efficiency or cell viability. Therefore, we focused on genes with higher expression rather than those with low expression by removing low-quality cells. Next, the accuracy of each scRNA-seq data depends on the experience of each facility. Our integrated scRNA-seq results may demonstrate selection bias although all datasets have already been published thus far and a certain level of reliability has been ensured (page 13, lines 377–382 and page 14, lines 383–385). Evaluating accurate gene expression levels at each cell between the HER2 subtype and the Luminal subtype is not the main objective of our investigation. We mainly aimed to provide novel insights on what should be clinically considered in treating HER2 breast cancer by capturing characteristics of heterogeneity in HER2 breast cancer compared with luminal breast cancer.

By employing bioinformatics tools, cells are classified into different clusters. However, the characteristics of these classifications, their implications for breast cancer management, how to selectively choose treatment approaches, and their effects on patient prognosis should be prominently discussed. This discussion should go beyond the presentation of data, as it holds significant importance.

A. We added our recommendation for breast cancer management and its effects on patient prognosis in the Discussion section (page 14, lines 386–402). First, our results identified intra-tumoral heterogeneity, but inter-patient heterogeneity could also be important along with that issue, as shown in Figure 4. Therefore, clinicians need to consider medical treatment not only focusing on the genetic differences in each tumor cell but also developing treatment following the genetic characteristics of each patient in the future. Next, the distribution of ERBB2 expression significantly differed among three subtypes, including pure HER2, Luminal-HER2, and Luminal subtypes. ERBB2 expression was diffusely distributed on all clusters in the pure HER2 subtype. Meanwhile, ERBB2 expression was localized in a specified cluster. The result indicates that other clusters with no ERBB2 expression could have resulted in the absence of response for HER2-targeted therapy. Clinicians may need to carefully consider treatment strategies for breast cancer with Luminal-HER2 subtypes rather than those with pure-HER2 subtypes. Interestingly, a slight ERBB2 expression was found in the Luminal subtype, and the results were compatible with the previous results on clinical trials that ERBB2 targeted therapy demonstrated better survival in patients with HER2-low breast cancer.

In addition, there are several issues that need to be addressed:

  1. Figure 2D lacks MKI67 plots.

A1. We added MKI67 plots in Figure 2D. A high expression of MKI67 was observed in cluster 5, as described in the manuscript (page 5, lines 178–179).

  1. It is necessary to provide clear definitions for the Luminal subtype and HER2 subtype.

A2. We added clear definitions of both the Luminal and HER2 subtypes in the Methods section (page 2, lines 88–96).

Luminal subtype: Cases with either ER or PgR positivity on IHC staining. ER and PgR status were considered positive using a cut-off value of ≥1% according to the American Society of Clinical Oncology/College American Pathologists (ASCO/CAP) guideline (referene number: 9). We considered either 2+ or 3+ as positive in datasets which were described as 0+, 1+, 2+, or 3+.

HER2 subtype: Cases with HER2 positivity. We considered 3+ or 2+ with HER2 amplification on IHC staining as positive with reference to the ASCO/CAP guideline (reference number: 10). The results of HER2 amplification is necessary in cases with unavailable HER2 results on IHC score.

  1. In Figure 3B, cluster 3 and 7 were selected for comparison. However, what about the diffusion across other clusters? In clinical practice, breast cancer is categorized into Luminal A, Luminal B, HER2-positive, and triple-negative subtypes. It's important to define the Luminal and HER2 groups and provide an explanation for their selection.

A3. Other clusters, except for cluster 3, demonstrated no significant differences (page 5, lines 184–185). Cluster 7 was marginally different, as described in our results.

As you pointed out, breast cancer is mainly categorized into the Luminal, HER2-positive, and triple-negative subtypes. We selected the Luminal subtype as a reference to investigate tumor heterogeneity of HER2-positive breast cancer because cross-talk between ER and HER2 has already been reported thus far (reference number: 8). We aimed to focus on the comparison between the Luminal and HER2 subtypes to investigate the relationship between ER and HER2 status. We added this explanation in the Introduction section (page 2, lines 70–72).

  1. GO analysis was performed for each cluster. By examining the specific differences in GO terms within each group, what cellular characteristics do these differences indicate, and what insights do these data provide?

A4. As our description for GO analysis was poor, we added some sentences and explained the data more appropriately as shown below (page 5, lines 190–193). Hope this explanation will be satisfactory for you.

“Then, based on the clusters, we performed pathway enrichment analysis to identify cluster-specific function and signatures (Figure 3C). The function of cluster 1 was identified as correlation with estrogen-dependent gene expression. Cluster 5 was associated with cell proliferation which were compatible with high MKI67 expression in the cluster analysis. Meanwhile, cluster 7, which were up regulated in HER2 subtype, were associated with cell adhesion. Clusters 3 and 9 might be associated with immune response based on GO terms. Further, clusters 2, 4, 6, and 10 contained similar GO terms, such as peptide chain elongation and eukaryotic translation elongation; thus, they could be similar cell populations, although they were classified as different clusters.”

  1. The criteria for selecting EMT markers and CSC markers need to be clarified。

A5. Representative EMT markers and CSC markers were selected in Figure 4. We added some references to select those markers as follows (page 3, lines 140–141).

  1. There seems to be inconsistency in cluster numbers. Figure 3A shows 11 clusters, while Figure 4 indicates that the HER2 group has 14 clusters and the Luminal group has 13 clusters.

A6. Figure 3A shows the results of clustering all data sets that correspond to either HER2 subtype or Luminal subtype. Hence, 11 clusters were revealed in the UMAP plots. Meanwhile, Figure 4 indicates the average gene expression at each case. We collected 14 HER2 subtype cases and 13 Luminal subtype cases in the all integrated data sets. Supplementary Table 2 shows the details of each case (Line A) (e.g., Cases of “HER2_1” [The cell in column A of row 3] corresponds to “CID3586” [The cell in column B of row 3].)

Round 2

Reviewer 2 Report

They addressed all questions well.